# Melatonin Improves the Fertilization Capacity of Sex-Sorted Bull Sperm by Inhibiting Apoptosis and Increasing Fertilization Capacitation via MT1

**DOI:** 10.3390/ijms20163921

**Published:** 2019-08-12

**Authors:** Chong-Yang Li, Hai-Sheng Hao, Ya-Han Zhao, Pei-Pei Zhang, Hao-Yu Wang, Yun-Wei Pang, Wei-Hua Du, Shan-Jiang Zhao, Yan Liu, Jin-Ming Huang, Jing-Jing Wang, Wei-Min Ruan, Tong Hao, Russel J. Reiter, Hua-Bin Zhu, Xue-Ming Zhao

**Affiliations:** 1Embryo Biotechnology and Reproduction Laboratory and the Center of Domestic Animal Reproduction & Breeding, Institute of Animal Science (IAS), Chinese Academy of Agricultural Sciences (CAAS), Beijing 100193, China; 2Dairy Cattle Research Center, Shandong Academy of Agricultural Sciences, Jinan 250131, China; 3International Joint Center for Biomedical Innovation, School of Life Sciences, Henan University, Kaifeng 475004, China; 4Department of Cell Systems and Anatomy, The University of Texas Health Science Center at San Antonio (UT Health), San Antonio, TX 78229, USA

**Keywords:** melatonin, fertilization, sex-sorted, sperm, bull

## Abstract

Little information is available regarding the effect of melatonin on the quality and fertilization capability of sex-sorted bull sperm, and even less about the associated mechanism. Sex-sorted sperm from three individual bulls were washed twice in wash medium and incubated in a fertilization medium for 1.5 h, and each was supplemented with melatonin (0, 10^−3^ M, 10^−5^ M, 10^−7^ M, and 10^−9^ M). The reactive oxygen species (ROS) and endogenous antioxidant activity (glutathione peroxidase (GPx); superoxide dismutase (SOD); catalase (CAT)), apoptosis (phosphatidylserine [PS] externalization; mitochondrial membrane potential (Δψm)), acrosomal integrity events (malondialdehyde (MDA) level; acrosomal integrity), capacitation (calcium ion [Ca^2+^]_i_ level; cyclic adenosine monophosphate (cAMP); capacitation level), and fertilization ability of the sperm were assessed. Melatonin receptor 1 (MT1) and 2 (MT2) expression were examined to investigate the involvement of melatonin receptors on sex-sorted bull sperm capacitation. Our results show that treatment with 10^−5^ M melatonin significantly decreased the ROS level and increased the GPx, SOD, and CAT activities of sex-sorted bull sperm, which inhibited PS externalization and MDA levels, and improved Δψm, acrosomal integrity, and fertilization ability. Further experiments showed that melatonin regulates sperm capacitation via MT1. These findings contribute to improving the fertilization capacity of sex-sorted bull sperm and exploring the associated mechanism.

## 1. Introduction

Flow cytometry has been widely used to sort sperm based on the difference in DNA content of X and Y spermatozoa. High quality frozen sex-sorted sperm plays an important role in the prevention and control of sex-linked genetic diseases, factory production of sexed embryos, and shortening the animal breeding cycle [1]. However, the preparation of sex-sorted frozen sperm involves the processes of attenuation, dying, laser light, freezing and thawing, all of which produce a large number of reactive oxygen species (ROS). Several studies have shown that the low quality and fertilization capability of sex-sorted sperm are closely associated with an excessive level of ROS [2,3,4].

Intracellular ROS is primarily derived from the mitochondria in the electron transport chain through the oxidative phosphorylation pathway [5], and sometimes important second messenger required to regulate cellular function [6]. Cells use a variety of antioxidant enzymes to remove excess ROS to reduce them to a steady state. Oxidative stress, however, disrupts the interior balance of the sperm’s antioxidant defense system, and induces oxidative injury to sperm [7].

The cell membranes of sperm are rich in polyunsaturated fatty acids (PUFAs) [8], the later are sensitive to ROS and finally result in lipid peroxidation (LPO), which is a chain reaction that produces cytotoxic aldehydes [9] (e.g., MDA). Moreover, MDA can bind to proteins and induce oxidative modification of crucial cell proteins [9], which negatively influences mitochondrial function and causes damage to the midpiece and axonemal structure of sperm [10]. In addition, increased LPO and altered membrane functional can render sperm dysfunctional through impaired metabolism, motility, acrosome reaction, reactivity, and fusogenic capacity, as well as oxidative damage to the sperm DNA [11], these changes together contribute to a low fertilization capacity of sperm.

Melatonin is an endogenous substance produced by the pineal gland that exhibits unique anti-oxidative [12], anti-inflammatory [13] and even anti-cancer [14] functions. Moreover, melatonin [15,16,17,18] and its metabolic products [19] are also effective antioxidants, and the antioxidant power of melatonin is significantly greater than that of vitamin E or C and 5 to 15 times greater than that of glutathione [20]. To date, several studies have reported that melatonin can protect the sperm viability of non-sorted frozen sperm in mouse [21], stallion [22], ram [23,24], bull [25], and human [26]. However, little information is still available regarding the effect of melatonin on sex-sorted bull sperm, and even less about the associated mechanisms.

Considering the lack of information in this area, sex-sorted sperm from three individual bulls were washed in wash medium and incubated in fertilization medium for 1.5 h, both of which were supplemented with five different concentrations of melatonin (0, 10^−3^ M, 10^−5^ M, 10^−7^ M, and 10^−9^ M). ROS levels and endogenous antioxidant activities (glutathione peroxidase (GPx); superoxide dismutase (SOD); catalase (CA)), apoptotic events (phosphatidylserine (PS) externalization; mitochondrial membrane potential (Δψm)), acrosomal integrity events (malondialdehyde (MDA) level; acrosomal integrity), capacitation events (calcium ion [Ca^2+^]_i_ level; cyclic adenosine monophosphate (cAMP); capacitation level), and fertilization ability of the sperm were examined. To investigate the involvement of melatonin receptors on the fertilization capacity of sex-sorted bull sperm, the expression levels of melatonin receptor 1 (MT1) and 2 (MT2) were assessed and 4-P-PDOT (selective MT2 antagonist) or luzindole (nonselective MT1/MT2 antagonist) was supplemented in the wash and fertilization medium with melatonin. The findings contribute to information that should improve the fertilization capacity of sex-sorted bull sperm and also explore the associated mechanisms.

## 2. Results

### 2.1. Effect of Melatonin on ROS Level and CAT, GPx, and SOD Activities in Sex-Sorted Bull Sperm

Figure 1A shows a representative image of ROS staining of sex-sorted bull sperm. For each bull, the ROS level of every group of melatonin-treated sperm was significantly lower than that of the control group (*p* < 0.05). The ROS level of the 10^−3^ M melatonin group was the lowest among all groups (*p* < 0.05), as shown in Figure 1B. As shown in Figure 1C, the CAT level of the 10^−5^ M group (9.9 U/mgprot), 10^−3^ M group (8.5 U/mgprot), and 10^−7^ M group (7.6 U/mgprot) was significantly higher than that of the control group (6.1 U/mgprot; *p* < 0.05), with the 10^−5^ M melatonin group exhibiting the highest value.

The GPx level of the 10^−5^ M group (148.6 U/mgprot), 10^−3^ M group (136.5 U/mgprot), and 10^−7^ M group (131.5 U/mgprot) was significantly higher than that of the control group (118.0 U/mgprot; *p* < 0.05), with 10^−5^ M melatonin group exhibiting the highest value.

The SOD level of 10^−5^ M group (16.9 U/mgprot), 10^−3^ M group (13.6 U/mgprot), 10^−7^ M group (13.2 U/mgprot), 10^−9^ M group (12.1 U/mgprot) was significantly higher than that of the control group (11.3 U/mgprot; *p* < 0.05), with 10^−5^ M melatonin group exhibiting the highest value.

### 2.2. Effect of Melatonin on the PS Translocation Events of Sex-Sorted Bull Sperm

Figure 2A–D shows representative images of PS analysis of sex-sorted bull sperm.

As shown in Figure 2E, the viable sperm of 10^−5^ M group (46.9%) and 10^−3^ M group (31.6%) was significantly higher than that of the control group (23.2%; *p* < 0.05), and the late apoptotic sperm of 10^−5^ M group (31.2%) and 10^−3^ M group (32.3%) was significantly lower than that of the control group (52.8%; *p* < 0.05).

### 2.3. Effect of Melatonin on the Δψm Levels of Sex-Sorted Bull Sperm

Figure 3A−C shows the representative images of the Δψm analysis of sex-sorted bull sperm.

As shown in Figure 3D, the percentage of sperm with high Δψm of the 10^−3^ M, 10^−5^ M, and 10^−7^ M group (18.7–26.5%) was significantly higher than that of the control group (14.00%; *p* < 0.05), with 10^−5^ M melatonin group (26.5%) exhibiting the highest value. Moreover, the percentage of sperm with low Δψm of the 10^−5^ M group (71.7%) was significantly lower than for the other groups (78.5–85.4%; *p* < 0.05).

### 2.4. Effect of Melatonin on the Levels of MDA in Sex-Sorted Bull Sperm

As shown in Figure 4, the level of MDA of 10^−3^M (7.0 nM), 10^−5^ M (4.5 nM) and 10^−7^ M (6.9 nM) groups were significantly lower than that of the control group (11.7 nM; *p* < 0.05).

### 2.5. Effect of Melatonin on the Acrosome Integrity of Sex-Sorted Bull Sperm

Figure 5A−D shows representative images of an acrosome integrity analysis of sex-sorted bull sperm. As shown in Figure 5E, the percentage of viable sperm with intact acrosomes (FITC-PNA−/PI-) of 10^−3^ M (52.7%) and 10^−5^ M (68.5%) melatonin groups was significantly higher than those of the control group (41.5%; *p* < 0.05).

The percentage of dead sperm with an intact acrosome (FITC-PNA-/PI+) of 10^−5^ M (17.7%) melatonin group was significantly lower than that of other groups (24.5–25.8%, respectively).

### 2.6. Effect of Melatonin on the [Ca^2+^]_i_ and cAMP Levels in Sex-Sorted Bull Sperm

As shown in Figure 6A, the [Ca^2+^]_i_ level of 10^−3^ M, 10^−5^ M, and 10^−7^ M melatonin group was significantly higher than that of control group (*p* < 0.05). The [Ca^2+^]_i_ level of the 10^−5^ M melatonin group was the highest among all groups (*p* < 0.05).

As shown in Figure 6B, the level of cAMP of the 10^−3^ M (34.5 nM), 10^−5^ M (57.0 nM), and 10^−7^ M (33.0 nM) melatonin group was significantly higher than that of the control (21.8 nM) and the 10^−9^ M melatonin group (23.8 nM; *p* < 0.05). The level of cAMP of the 10^−5^ M melatonin group was highest among all groups (*p* < 0.05).

### 2.7. Effect of Melatonin on the Capacitation of Sex-Sorted Bull Sperm

Figure 7A−E shows representative images of the capacitation analysis of sex-sorted bull sperm.

As shown in Figure 7F, the percentage of live sperm with capacitation (YoPro^−^/M540^+^) of 10^−3^ M (46.6%), 10^−5^ M (58.8%), and 10^−7^ M (42.8%) melatonin groups was significantly higher than that of the control groups (22.5%, *p* < 0.05), with 10^−5^ M melatonin groups exhibiting the highest values.

The percentage of dead sperm without capacitation (YoPro^+^/M540^−^) of the 10^−3^ M (44.1%), 10^−5^ M (31.3%), and 10^−7^ M (48.6%) melatonin-treated sperm was significantly lower than that of the control group (68.3%; *p* < 0.05), with 10^−5^ M melatonin groups displaying the lowest values.

Moreover, the percentage of live sperm without capacitation (YoPro^−^/M540^−^) of melatonin groups (7.0–7.8%) were similar to that of the control group (7.7%; *p* > 0.05).

### 2.8. Effect of 4-P-POD and Luzindole on the Levels of [Ca^2+^]_i_, cAMP, and Capacitation of Sex-Sorted Bull Sperm

As shown in Figure 8A, both MT1 and MT2 were expressed in sex-sorted bull sperm.

As shown in Figure 8B, the level of [Ca^2+^]_i_ of the luzindole + 10^−5^ M melatonin group was significantly lower than those of the 10^−5^ M melatonin and 4-P-PDOT + 10^−5^ M melatonin groups, but were higher than that of the control group (*p* < 0.05).

As shown in Figure 8C, the level of cAMP of the 4-P-PDOT + 10^−5^ M melatonin group (55.2 nM) was similar to that of the 10^−5^ M melatonin group (55.7 nM) and significantly higher than that of control group (18.1 nM) and luzindole + 10^−5^ M melatonin group (43.9 nM).

As shown in Figure 8D, the percentages of viable sperm with capacitation (YoPro^−^/M540^+^) of luzindole + 10^−5^ M melatonin group (45.7%) was significantly lower than those of 10^−5^ M melatonin (58.7%) and 4-P-PDOT + 10^−5^ M melatonin groups (57.8%; *p* < 0.05); however, the level was significantly higher than that of the control group (20.7%; *p* < 0.05).

### 2.9. Effect of Melatonin on IVF Efficiency in Sex-Sorted Bull Sperm

As shown in Figure 9, the cleavage percentage of 10^−5^ M group (57.6%) was significantly lower than that of un-sorted group (78.0%; *p* < 0.05), but higher than that of other groups (27.9–39.1%; *p* < 0.05). The blastocyst percentage of 10^−5^ M group (30.0%) was similar to that of un-sorted group (34.0%), and higher than that of control group (13.5%; *p* < 0.05).

## 3. Discussion

Previous studies involving stallions [22], rams [23,24], bulls [25], and humans [26,27] have demonstrated that melatonin efficiently removes excess ROS from sperm. Our results also showed that the level of ROS was significantly lower in all of the melatonin-treated groups compared to the controls (Figure 1B), indicating that ROS was removed efficiently from these groups. As previously reported, both melatonin and its metabolites are powerful antioxidants [12], which contribute to explaining these results.

Melatonin reportedly increases the mRNA expression levels of *CAT* and *GPX* in bovine oocytes [28] and mouse embryos [29], SOD activity in wistar rat plasma [30] and rat cardiomyocytes [31]. As shown in Figure 1C, our results showed that melatonin treatment significantly increased the activities of CAT, GPx, and SOD. Collectively, this evidence indicates that melatonin significantly enhances the activities of endogenous antioxidants, which also helps to explain the lower level of ROS in the melatonin-treated group (Figure 1B). Moreover, melatonin regulates the expression and activity of antioxidants by the melatonin receptor and calcium-calmodulin complex [15], which explains the increased CAT, GPx, and SOD activities in melatonin-treated bull sperm (Figure 1C).

Similar to the reports involving non-sorted sperm in rams [23] and humans [32], our results showed that 10^−5^ M melatonin significantly increased the level of live sperm and reduced early necrotic sperm (Figure 2). Moreover, we found that treatment with 10^−3^ M, 10^−5^ M, and 10^−7^ M melatonin could significantly improve Δψm (Figure 3) in the sex-sorted sperm. Oxidative stress has been proven to decrease Δψm, resulting in increased apoptosis in bovine oocytes [33] and human sperm [34,35]. Therefore, lower levels of ROS (Figure 1B), higher levels of endogenous antioxidants (Figure 1C) in the melatonin groups and the capability of melatonin and its metabolities in directly repairing oxidized DNA [36] explained the lower percentage of early necrotic sperm (Figure 2) and higher Δψm sperm percentage (Figure 3).

MDA, an end-product of LPO, causes serious harm to the sperm membranes, and results in reduced function and structure of the sperm [37]. Similar to the results obtained from rat cardiomyocytes [31], mouse ovaries [38], and unsorted human sperm [39], we found that treatment with 10^−3^ M, 10^−5^ M, and 10^−7^ M melatonin significantly decreased the level of MDA in the bull sperm (Figure 4). Since ROS can induce LPO of PUFAs in the sperm membrane which results in increased levels of MDA [9], the lower levels of ROS in the melatonin groups explained the lower levels of MDA.

An acrosome reaction always occurs during the process of the sperm and oocyte combination, and a damaged acrosome results in the loss of fertility [40]. Similar to the results observed in unsorted bull [41] and boar [42] sperm, our experiments showed that melatonin protected the acrosome integrity of viable sex-sorted bull sperm (Figure 5). It has been demonstrated that ROS stress damages the acrosome integrity [43] since ROS stress leads to increased levels of MDA [9], which damages the sperm acrosome [37]. Therefore, the lower levels of ROS (Figure 1C) and MDA (Figure 4) in the melatonin-treated sperm explained the higher acrosome integrity in the melatonin groups (Figure 5).

[Ca^2+^]_i_ efflux is an important event required for membrane phospholipids at the sperm head that leads to capacitation. cAMP is a second messenger that is essential for many biological processes, including many sperm functions known to be regulated by cAMP [44]. Herein, we found that treatment with 10^−5^ M melatonin could significantly improve the levels of [Ca^2+^]_i_ (Figure 6A) and cAMP (Figure 6B) in sex-sorted bull sperm. Melatonin has been shown to increase [Ca^2+^]_i_ levels by promoting the combination of the IP3 and IP3 receptor located on the endoplasmic reticulum and cAMP by inhibiting adenylate cyclase activity via membrane-bound G protein-coupled melatonin receptors, MT1 and MT2 [45], which explains the increased levels of [Ca^2+^]_i_ (Figure 6A) and cAMP (Figure 6B) in the melatonin-treated groups.

Capacitation correlated with the fertilization capacity of sperm was firstly described more than 60 years ago [46]. Sperm capacitation is a complex process that involves calcium movement [47] and activation of cAMP-dependent pathways [48] that leads to the acrosome reaction and acquisition of the ability to fertilize the oocyte. As shown in Figure 7, our results showed that treatment with 10^−3^ M, 10^−5^ M, and 10^−7^ M melatonin significantly increased the percentage of viable sperm with capacity. It has been reported that the decrease in ROS [22] and increase in cAMP [49] and [Ca^2+^]_i_ levels [50] can promote the capacitation of sperm, which explained the higher capacitation level of sperm in the present study.

As shown in Figure 8A, we found that both the melatonin receptors, MT1 and MT2, are expressed in frozen sex-sorted bull sperm by Western blot. The melatonin receptors, MT1 and MT2, are variably expressed and located in the sperm of domestic animals. Although both MT1 and MT2 are expressed in ejaculated ram spermatozoa [51], no presence of MT1 or MT2 is found in stallion sperm [22]. In comparison, MT1 is distributed in the sperm head and flagellum of bulls, whereas MT2 is distributed in the neck of bull sperm [52]. Moreover, the protein level of MT2 is similar to that of MT1 in sheep oocytes at the GV stage, but lower than that of MT1 in sheep granulosa and cumulus cells at the GV stage, and higher than that of MT1 in sheep oocytes and cumulus cells at the MII stages [53]. The specific localization of MT1 and MT2 in cells is suggestive of their specific physiological functionality [54].

Until now, no conclusion could be drawn on the melatonin receptor subtype mediating the effects of melatonin on sleep [55]. However, there are many studies have shown that the vast majority of functional melatonin receptors are MT1, rather than MT2, in rat spinal cord cells [56] and mouse ovary [57], C6 astroglial cells [58], bovine embryos [59] and oocytes [28], and hamster sperm [60]. Similar to these studies, our results showed that the levels of [Ca^2+^]_i_, cAMP and capacitation of luzindole + 10^−5^ M melatonin groups were significantly lower than those of 10^−5^ M and 4-P-PDOT + 10^−5^ M melatonin groups (Figure 8B−D), confirming that the melatonin receptor, MT1, plays an important role in the regulation of levels of [Ca^2+^]_i_ and cAMP in sex-sorted bull sperm. Additionally, the levels of [Ca^2+^]_i_, cAMP, and capacitation in the luzindole + 10^−5^ M melatonin groups (Figure 8B−D) were significantly higher than that in the control groups, indicating that melatonin also regulates the level of [Ca^2+^]_i_ and cAMP in sex-sorted bull sperm using other approaches, which requires further research in the future.

As shown in Figure 9, treatment with 10^−5^ M melatonin significantly improved the fertilization capacity of sex-sorted bull semen and the capacity for embryo development, consisted with previous studies in non-sorted ram sperm [23]. The capacitation and acrosome reaction of sperm are important prerequisites for fertilization [61]. In addition, apoptotic-like changes [23], PS translocation, and low Δψm [62] are negatively correlated with the fertilizing capacity of sperm. Lower PS translocation levels (Figure 2), higher levels of Δψm (Figure 3), acrosome integrity (Figure 5), and the capacitation (Figure 7) of melatonin-treated sperm explained their higher fertilization capacity (Figure 9).

The findings of the present study indicated that treatment with 10^−5^ M melatonin significantly decreased the level of ROS and increased the activities of GPx, SOD, and CAT in the sex-sorted bull sperm, which further inhibited PS externalization and the level of MDA; additionally, melatonin improved Δψm, acrosomal integrity, and fertilization ability. Furthermore, the melatonin receptor, MT1, was involved in the regulation of sperm capacitation by melatonin. These findings contribute to information on a treatment that improves the fertilization capacity of sex-sorted bull sperm and explores the associated mechanisms.

## 4. Materials and Methods

### 4.1. Materials

Unless specifically stated, all chemicals used in this study were purchased from Sigma-Aldrich Chemical Company (St. Louis, MO, USA) and plastics items were purchased from Corning Company (Corning, NY, USA).

### 4.2. The Treatment of Sex-Sorted Semen

Frozen bovine sex-sorted semen was provided by OX biotechnology Ltd. (Shandong, China), the samples were thawed, added to 15-mL conical centrifuge tubes containing 3 mL of wash medium (Brackett and Oliphant medium supplemented with 2.5 mM caffeine [63], and washed twice via centrifugation at 328× *g* for 5 min. The semen pellets were re-suspended in fertilization medium (BO medium with 20 mg/mL BSA, 100 IU/mL penicillin, 100 μg/mL streptomycin, and 20 μg/mL heparin) for a final concentration of 1 × 10^6^/mL. The re-suspended semen was incubated for 1.5 h at 38.5 °C in a 5% CO_2_ humidified atmosphere. The semen was then subsequently used for the following assays.

### 4.3. Analysis of the Level of ROS and Endogenous Antioxidant Activity

The ROS staining procedure was performed according the method described by Zhao et al. [33], in which each group of semen was stained with 10 μM 2′,7′- dichlorodihydrofluorescein diacetate (H_2_DCFDA; Molecular Probes, Eugene, OR, USA) in the dark at 38.5 °C for 20 min. The sperm were washed twice by centrifugation at 328× *g* for 5 min, the sperm pellet was re-suspended in PBS, and analyzed using a Laser confocal microscope (MoFlo XDP, Bekeman, CA, USA) to observe the level of intracellular ROS.

To analyze the level of endogenous antioxidant activity, each group of sperm underwent ultrasonic destruction and 1000× *g* centrifugation for 5 min and the supernatants were collected to detect the level of antioxidant enzyme activities (GPx, SOD, and CAT). The commercial kits used to detect the enzyme activities of these three antioxidants were measured using a Jiancheng commercial kit (Nanjing Jiancheng Bioengineering Institute, Nanjing).

The GPx enzymatic activity was assessed by the catalytic reaction speed between glutathione (GSH) and H_2_O_2_, as the GSH present in the samples reacts with DTNB (5,5-dithiobis- 2-nitrobenzoic acid), creating a yellow product that is quantified at 412 nm using a microplate reader (Tecan Group Ltd., Tecan Infinite 200 Pro, Männedorf, Switzerland). One unit of GPx enzymatic activity is equivalent to 100 µmol/L GSH reduced per minute per milligram (mg) protein after excluding the effect of a non-enzymatic-catalyzed reaction. The results are expressed as U/mg protein.

The level of SOD activity was measured by the degree of inhibition using xanthine and a xanthine oxidase system to generate superoxide radicals (O_2_^•−^), which will react with hydroxylamine to form a purplish red product that can be quantified at 550 nm using the microplate reader (Tecan Group Ltd., Männedorf, Switzerland). The results are expressed as U/mg protein.

CAT activity was measured based on H_2_O_2_ consumption, and the remaining H_2_O_2_ reacts with ammonium molybdate to form a yellow product that can be quantified at 405 nm using a microplate reader (Tecan Group Ltd.). The results are expressed as U/mg protein.

### 4.4. Analysis of PS Translocation Events and Level of Δψm

The presence of PS translocation to the outer leaflet of the plasma membrane is indicative of cells in apoptosis, and can be detected using an Annexin V-FITC Apoptosis Kit (Biovision, Mountain View, CA, USA). Samples were treated with 1 × Annexin V binding buffer and washed twice via centrifugation at 328× *g* for 5 min. The sperm pellet was resuspended in 1 × Annexin V binding buffer, incubated in Annexin V-FITC and PI for 15 min in the dark at 38.5 °C, and was then evaluated by flow cytometry. At least 10,000 sperm were analyzed for each sample. The sperm were divided into four groups according to the staining results: viable sperm (Annexin V-FITC^−^ and PI^−^); early apoptotic sperm (Annexin V-FITC^+^ and PI^−^); early necrotic apoptotic sperm (Annexin V-FITC^+^ and PI^+^); and necrotic sperm (Annexin V-FITC^−^ and PI^+^) [64]. Moreover, unstained sperm and sperm stained with Annexin V-FITC or PI were used as controls.

Δψm was evaluated by JC-1, a specific probe (MitoProbe JC-1 assay kit; Invitrogen, Carlsbad, CA, USA) according to the methods described by Zhao et al. [62]. Sperm from each group were incubated with 2 μM JC-1 at 38.5 °C in the dark for 30 min before being analyzed by flow cytometry. At least 10,000 events were analyzed for each sample. As suggested by the experimental protocol, unstained sperm were used as the negative control and semen treated with ultraviolet light were used as a positive control.

### 4.5. Analysis of the Level of MDA and Acrosome Integrity of the Sperm

According to the instructions provided with the MDA kit (Nanjing Jiancheng Bioengineering Institute, Nanjing, China), MDA content was measured with 2-thiobarbituric acid (TBA) by monitoring the absorbance at 532 nm using a spectrophotometer. Briefly, each group of semen was mixed with 0.5% TBA and boiled for 80 min, was quickly cooled, and the resulting supernatant liquid was directly injected into the spectrophotometer after high speed centrifugation. The MDA content was expressed as nM.

Peanut agglutinin (PNA) binds specifically to the inner surface of the external membrane of the acrosome, labeling acrosome-damaged spermatozoa [65]. In this experiment, each group of semen was stained with 10 μg/mL FITC-PNA and 12 μM PI at 38.5 °C for 15 min. The sperm were washed twice with centrifugation at 328× *g* for 5 min, re-suspended by DPBS, and analyzed by flow cytometry to observe the status of the sperm acrosome. The sperms were divided into four populations: (1) FITC-PNA^−^/PI^−^ as alive with intact acrosomes; (2) FITC-PNA^+^/PI^−^ as alive with damaged acrosomes; (3) FITC-PNA^−^/PI^+^ as dead with intact acrosomes; and (4) FITC-PNA^+^/PI^+^ as dead with damaged acrosomes.

### 4.6. Analysis of the Level of [Ca^2+^]_i_, Intracellular cAMP, and Capacitation of Sperm

Each group of sex-sorted sperm were stained with 10 μM Fluo-3AM at 38.5 °C for 30 min. After washing twice with DPBS, the sperm pellet was re-suspended in DPBS and the fluorescence value of each single sperm sample was analyzed by flow cytometry. The median fluorescence intensity was detected by an argon laser at 488 nm, and at least 10,000 cells were collected for each sample. The median fluorescence intensity values were used to analyze the level of [Ca^2+^]_i_ in the sperm [66].

For cAMP level detection, according to the methods described by Deng et al. [67], a cAMP ELISA kit was performed in accordance with the manufacturer’s protocol (Nanjing Jiancheng Bioengineering Institute, Nanjing, China).

Briefly, each group of semen samples was incubated with 75 nM YoPro-1 and 2 μM M540 at 38.5 °C for 15 min. The samples were then washed twice by centrifugation at 328× *g* for 5 min, re-suspended in DPBS, and analyzed by flow cytometry to observe sperm capacitation. The semen was classified by labeling as follows: (1) viable cells with low plasma membrane scrambling (YoPro^−^/M540^−^); (2) viable cells with high plasma membrane scrambling (YoPro^−^/M540^+^); (3) apoptotic cells with low plasma membrane scrambling (YoPro^+^/M540^−^); and (4) apoptotic cells with high plasma membrane scrambling (YoPro^+^/M540^+^) [68]. Samples with viable sperm with high plasma membrane scrambling (YoPro^−^/M540^+^) indicated capacitation [69].

### 4.7. Western Blot Examination of Melatonin Receptors MT1 and MT2

A Western blot of melatonin receptors, MT1 and MT2, were performed according to the methods described by Kang et al. [70]. Briefly, the proteins were extracted from the sex-sorted sperm, subjected to 12% sodium dodecyl sulfate-polyacrylamide gel electrophoresis for 2.5 h at 300 mA, then transferred to PVP membranes. The membranes were blocked with 5% BSA in 0.5% Tween-20-PBS for 1 h, and then incubated overnight at 4 °C with an anti-MT1 primary antibody (sc-390328, 1:2000 dilution; Santa Cruz, Dallas, Texas), anti-MT2 primary antibody (sc-398788, 1:2000 dilution; Santa Cruz), and anti-β-tubulin (β-tubulin, YM3030, 1:5000 dilution; Immunoway, Plano, Texas, USA). The membranes were incubated with secondary antibodies (mouse, ab6789, 1:2000, Abcam, Cambridge, MA, USA) for 1 h at room temperature in the dark. Finally, the proteins bands were detected using an enhanced chemiluminescence (ECL) detection kit.

### 4.8. IVF Assay

Bovine ovaries from local abattoirs were transported within 2 h to the laboratory in 30–35 °C physiological saline. Cumulus-oocyte complexes (COCs) were aspirated from follicles 2–8 mm in diameter by vacuum suction, and only those with a homogenous cytoplasm and more than three complete layers of compact cumulus cells were selected for IVM. A total of 50 immature COCs were subjected to IVM in four-well plates containing 750 μL IVM medium per well, and then cultured at 38.5°C in a 5% CO_2_ humidified atmosphere for 22 h–24 h. The IVM medium composed of HEPES-buffered M199 (Gibco BRL; Grand Island, NY, USA) supplemented with 10 μg/mL estradiol, 10 μg/mL follicle-stimulating hormone, 10 μg/mL heparin, 10% (*v*/*v*) fetal bovine serum (FBS; Gibco BRL Division), and 10 μg/mL luteinizing hormone.

After 22–24 h IVM, the COCs were transferred into 0.1% (*w*/*v*) hyaluronidase for 30 s to remove cumulus cells. Only oocytes with a homogenous cytoplasm and first polar body were selected and transferred to the fertilization drops. After 16–18 h fertilization, presumed fertilized oocytes were transferred to CR1aa medium [71] containing 6 mg/mL BSA for 48 h. Then, cleavage embryos were cultured in CR1aa medium supplemented with 10% FBS for an additional five days with half of the medium replaced every 48 h.

### 4.9. Experimental Design

The sex-sorted bull sperm were washed twice in wash medium and incubated in fertilization medium for 1.5 h, each of which were supplemented with five different concentrations of melatonin (0, 10^−3^ M, 10^−5^ M, 10^−7^ M, and 10^−9^ M). The levels of ROS and endogenous antioxidant activities (GPx, SOD, and CAT), apoptotic events (PS externalization and Δψm), acrosomal integrity-related events (MDA level and acrosomal integrity), capacitation-related events (Ca^2+^ level; cAMP level, and capacitation level), and the fertilization ability of these sperm were examined. Moreover, the levels of MT1 and MT2 protein expression were examined and 4-P-PDOT (selective MT2 antagonist) or luzindole (nonselective MT1/MT2 antagonist) was supplemented in the wash and fertilization medium with melatonin to investigate the involvement of melatonin receptors in the capacitation of sex-sorted bull sperm.

### 4.10. Statistical Analysis

All experiments were repeated at least three times and the data were expressed as the mean ± S.E.M. All data were analyzed using one-way analysis of variance (ANOVA)with Statistical Analysis System software 8.0 (SAS Institute, Cary, NC, USA). The percentage data were subjected to arcsine transformation prior to statistical analysis. The values were considered as statistically significant differences when *p* < 0.05.

## Figures and Tables

**Figure 1 ijms-20-03921-f001:**
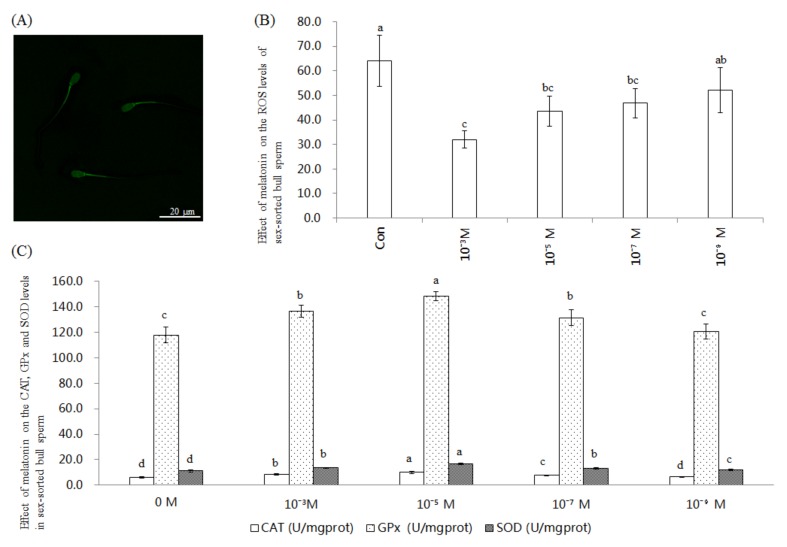
Effect of melatonin on ROS level and CAT, GPx, and SOD activities in sex-sorted bull sperm. (**A**) ROS staining. Scale bar = 20 μm. (**B**) ^a,b,c,d^ Values with no common superscripts represent statistical significance (*p* < 0.05), the same as below. (**C**) Effect of melatonin on the level of CAT, GPx, and SOD activities in sex-sorted bull sperm.

**Figure 2 ijms-20-03921-f002:**
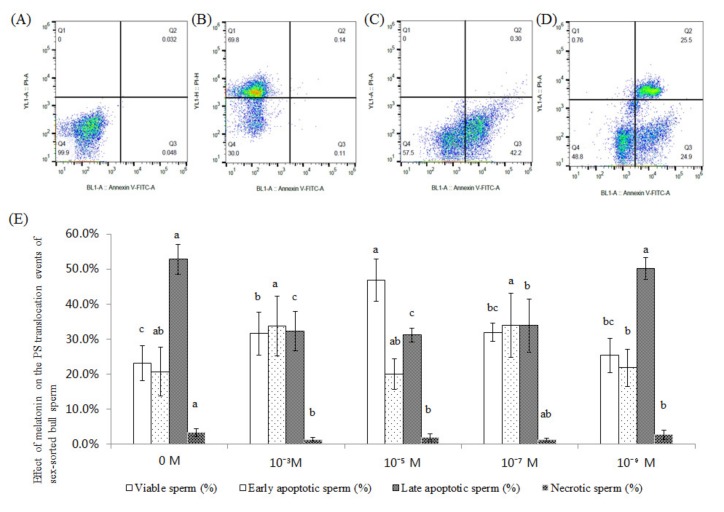
Effect of melatonin on the PS externalization events of sex-sorted bull sperm. (**A**) Negative control. (**B**) PI staining control. (**C**) Annexin V FITC staining control. (**D**) Analysis of sex-sorted bull sperm. Quadrants represented viable sperm (lower-left quadrant), early apoptotic sperm (lower-right quadrant), early necrotic sperm (upper-right quadrant), and necrotic sperm (upper-left quadrant). (**E**) Effect of melatonin on the PS externalization events of sex-sorted bull sperm. Lowercase letters represent statistical significant difference (*p* < 0.05).

**Figure 3 ijms-20-03921-f003:**
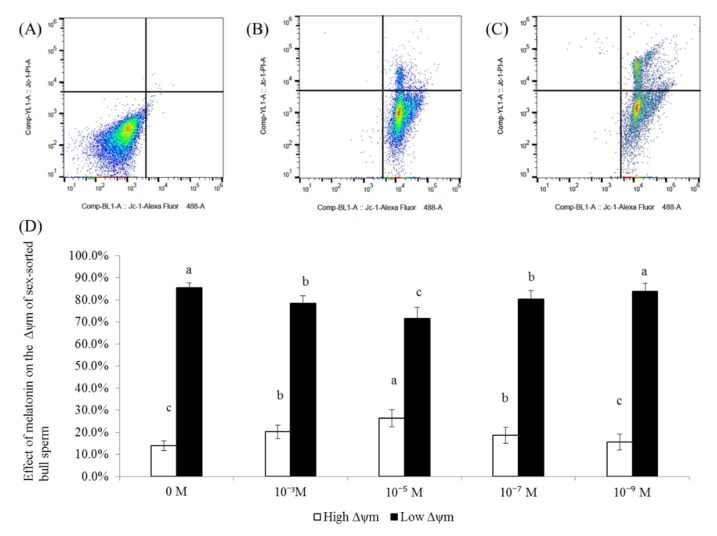
Effect of melatonin on the level of Δψm of sex-sorted bull sperm (**A**) Negative control. (**B**) Positive control. (**C**) Analysis of sex-sorted bull sperm. Quadrants represented low Δψm sperm (lower-right quadrant) and high Δψm sperm (upper-right quadrant). (**D**) Effect of melatonin on the level of Δψm of sex-sorted bull sperm. Lowercase letters represent statistical significant difference (*p* < 0.05).

**Figure 4 ijms-20-03921-f004:**
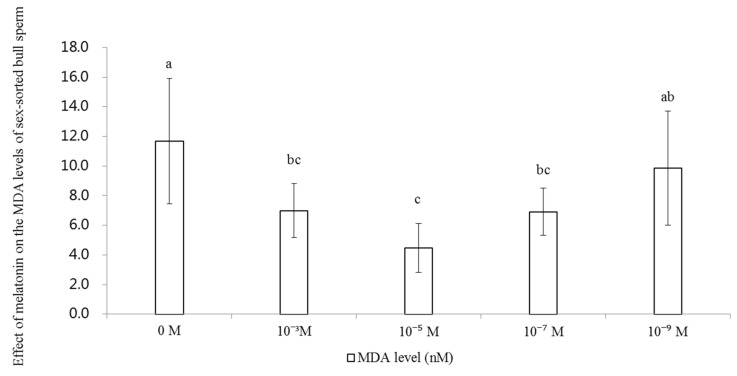
Effect of melatonin on the level of MDA of sex-sorted bull sperm. Lowercase letters represent statistical significant difference (*p* < 0.05).

**Figure 5 ijms-20-03921-f005:**
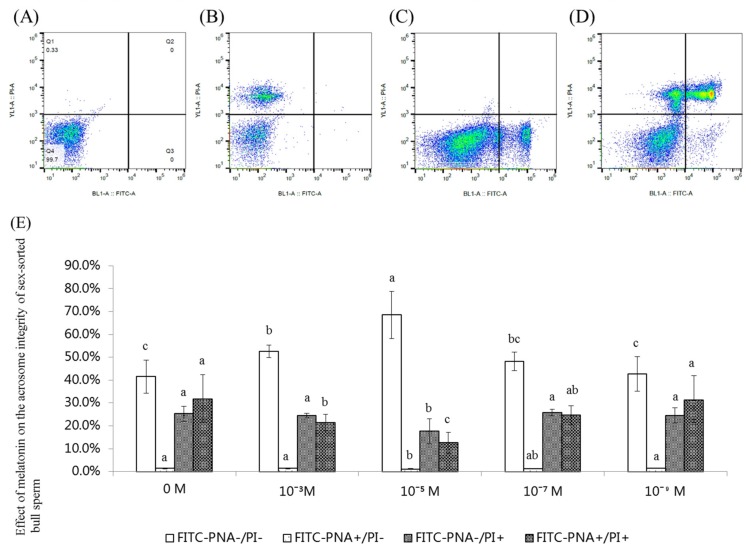
Effect of melatonin on the acrosome integrity of sex-sorted bull sperm. (**A**) Negative control. (**B**) PI staining. (**C**) FITC-PNA staining. (**D**) FITC-PNA/PI staining. Acrosome staining of sex-sorted sperm of bull. Quadrants represented viable sperm with integral acrosome (lower-left quadrant), viable sperm with damaged acrosome (lower-right quadrant), dead sperm with integral acrosome (upper-left quadrant), and dead sperm with damaged acrosomes (upper-right quadrant). (**E**) Effect of melatonin on the acrosome integrity of sex-sorted bull sperm. Lowercase letters represent statistical significant difference (*p* < 0.05).

**Figure 6 ijms-20-03921-f006:**
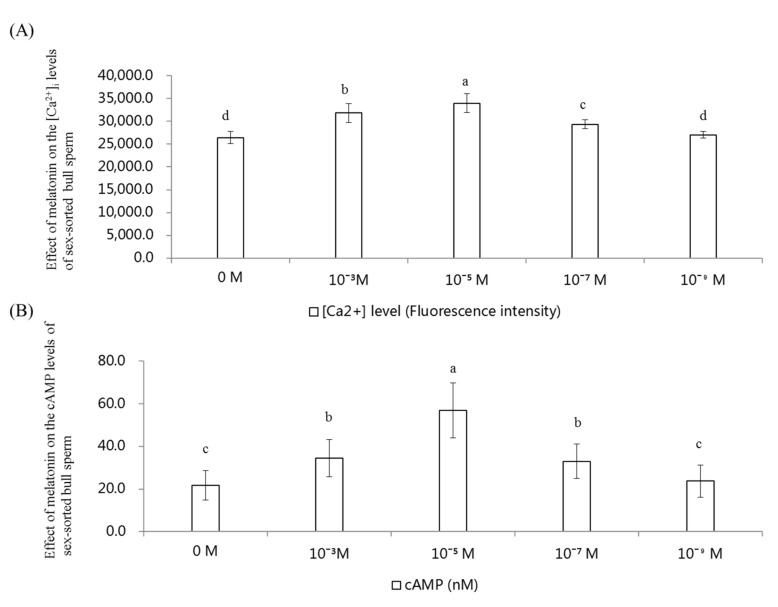
Effect of melatonin on the level of [Ca^2+^]_i_ and cAMP of sex-sorted bull sperm. (**A**) Effect of melatonin on the level of [Ca^2+^]_i_ of sex-sorted bull sperm. (**B**) Effect of melatonin on the level of cAMP of sex-sorted bull sperm. Lowercase letters represent statistical significant difference (*p* < 0.05).

**Figure 7 ijms-20-03921-f007:**
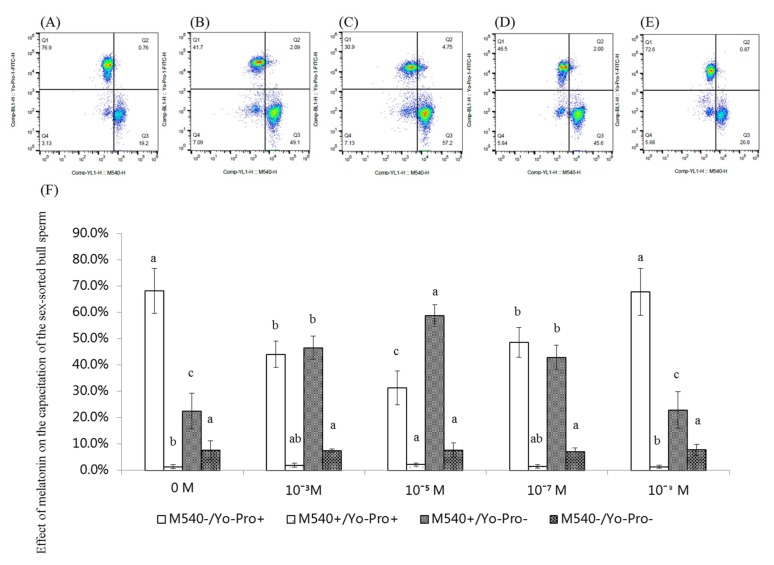
Effect of melatonin on sex-sorted bull sperm capacitation. (**A**) Representative image of 0 M group. (**B**) Representative image of 10^−3^ M group. (**C**) Representative image of 10^−5^ M sperm analysis. (**D**) Representative image of 10^−7^ M group. (**E**) Representative image of 10^−9^ M sperm analysis. The quadrants represented viable sperm without destabilized membranes (YoPro^−^ and M540^−^), viable sperm with destabilized membranes (YoPro^−^ and M540^+^), nonviable sperm without destabilized membranes (YoPro^−^ and M540^+^), and nonviable sperm with destabilized membranes (YoPro^+^ and M540^+^). (**F**) Effect of melatonin on sex-sorted bull sperm capacitation. Lowercase letters represent statistical significant difference (*p* < 0.05).

**Figure 8 ijms-20-03921-f008:**
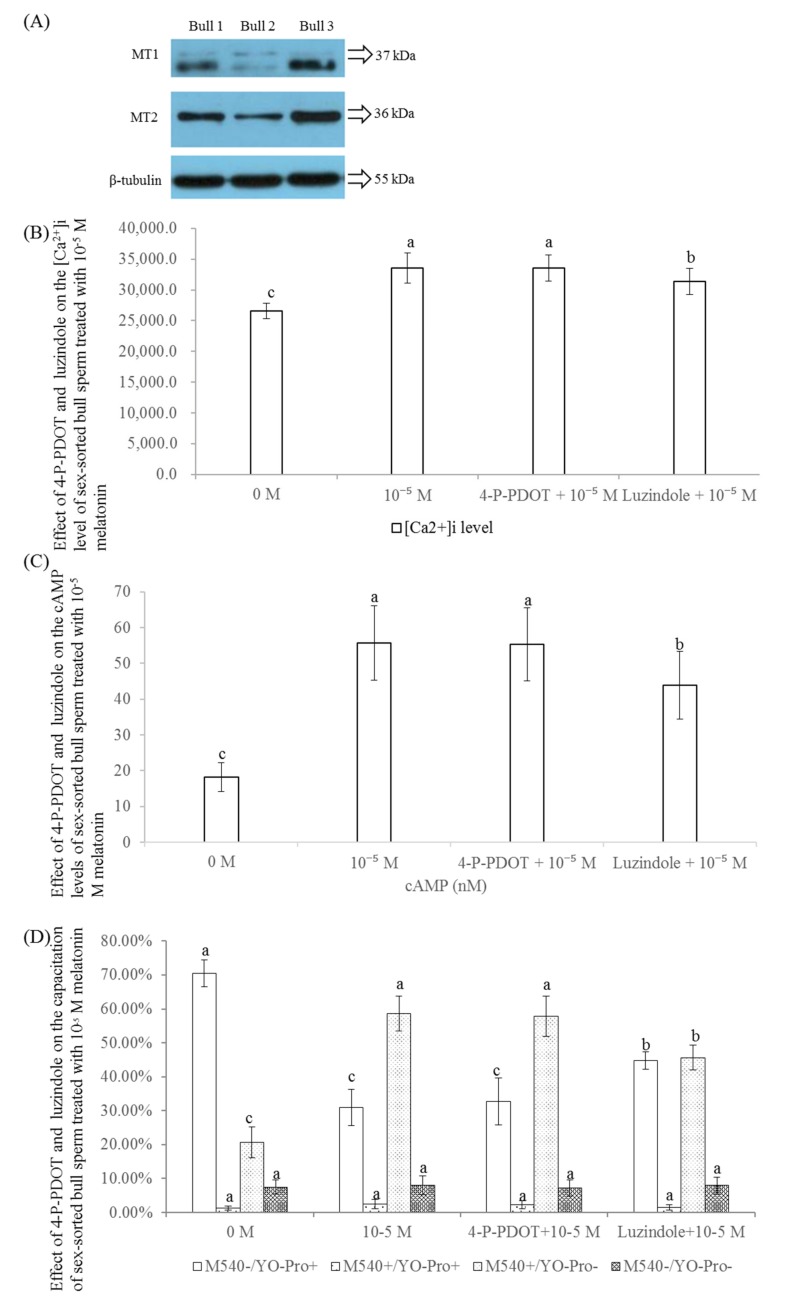
Effect of 4-P-POD and luzindole on the level of [Ca^2+^]_i_, cAMP, and capacitation of sex-sorted bull sperm. (**A**) Western blot image of MT1 and MT2 protein expression. (**B**) Effect of 4-P-POD and luzindole on the level of [Ca^2+^]_i_ of sex-sorted bull sperm. (**C**) Effect of 4-P-POD and luzindole on the level of cAMP of sex-sorted bull sperm. (**D**) Effect of 4-P-POD and luzindole on the level of capacitation of sex-sorted bull sperm. Lowercase letters represent statistical significant difference (*p* < 0.05).

**Figure 9 ijms-20-03921-f009:**
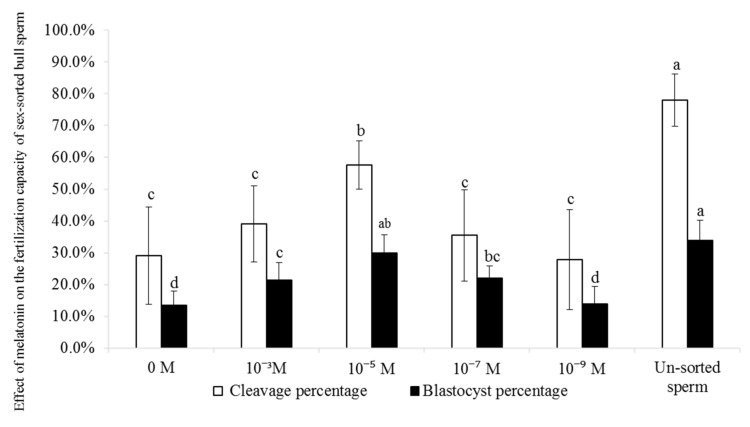
Effect of melatonin on the IVF efficiency of sex-sorted bull sperm. Lowercase letters represent statistical significant difference (*p* < 0.05).

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
