# Peer review of "Melatonin Improves the Fertilization Capacity of Sex-Sorted Bull Sperm by Inhibiting Apoptosis and Increasing Fertilization Capacitation via MT1"

_ijms, 2019, doi:10.3390/ijms20163921_

Round 1

Reviewer 1 Report

The article entitled „Melatonin improves 1 the fertilization capacity of sex sorted bull sperm by inhibiting apoptosis and increasing fertilization capacitation via MT1” is well written.

The abstract is informative and clear. The introduction is concise and shows the most important information to the understood purpose of the study, however, I would encourage the authors to add some paragraph/phrase regarding MT1 and MT2 inhibitors, to help the readers understood the idea of using it in this study.

Results I would change, especially the presentation of the data. Dividing the data for each bull separately makes the graph hard to read. It would be better to utilize the mean value of all 3 bulls. Also, the statistical analysis approach should be changed into repeated measures of ANOVA analysis.

In the references, I found one mistake – line 515 lack of italic in the journal name.

Overall I advise to accept the article after minor revision.

Author Response

However, I would encourage the authors to add some paragraph/phrase regarding MT1 and MT2 inhibitors, to help the readers understood the idea of using it in this study.

Response: Thanks for the question. The information of MT1 and MT2 inhibitors has been provided in Line79 and Line335-341. Actually, the information of these two chemicals are easy for readers to get on sigma website.

Results I would change, especially the presentation of the data. Dividing the data for each bull separately makes the graph hard to read. It would be better to utilize the mean value of all 3 bulls.

Response: Thanks for the question. We appreciated the suggestion of reviewer. From this experiment, we got to know that the response of sperm to the melatonin treatment depends on the individuals, so we prefer to present the data for each bull to show the differences between bulls. We hope that we can get the understanding of reviewer on this.

Also, the statistical analysis approach should be changed into repeated measures of ANOVA analysis.

Response: Thanks for the question. The method of statistical analysis was modified according to the comment of reviewer.

In the references, I found one mistake – line 515 lack of italic in the journal name.

Response: Thanks for the question. I have revised this content according to the comment of reviewer.

Reviewer 2 Report

The manuscript drafted from Yang Li et al., provided a comprehensive analysis regarding the melatonin effectiveness on the fertilization capacity of sex sorted bull sperm. 

Authors ' conclude that melatonin treatment improves the fertilization capacity of sex sorted bull sperm mainly through apoptosis inhibition.

The introduction provides sufficient background and includes all relevant references. Research design is appropriate and methods employed are adequately described. Results are clearly presented and fully support the respective conclusions reported. However, moderate English changes are required. In addition authors should consider to provide data regarding the number of samples analyzed in order to support the strength of the statistical analysis performed. This reviewer anticipates to see the revised version of this manuscript. 

Author Response

However, moderate English changes are required.

Response: Thanks for the question. Although this manuscript had been copyedited by a professional English editing service before submission, we still had it revised by a native English speaker to make it more fluent and readable according to the requirement of reviewer.

In addition authors should consider to provide data regarding the number of samples analyzed in order to support the strength of the statistical analysis performed.

Response: Thanks for the question. As mentioned in 4.1.1 Statistical analysis, all experiments were repeated at least three times. Meanwhile, details of population in samples were provided in the material and methods section. More importantly, it may be better to remain the original type of figures in order to avoid the complexity. We hope that we can get the understanding of reviewer on this.  

Round 2

Reviewer 1 Report

The authors correction on the manuscript have increased its readability. Despite I’m satisfied with two answers, I cannot understand the reasons for showing individual variations in results.

These differences did not influence the general trend in the figures: 1, 3, 4, 6, 7 and 8. Some trend disturbance is visible in figures 2 and 5. Questionable is figure 9 where data of fertilisation capacity shows the same trend but individual variation is seen in the blastocyst BAX and BCL2L1 expression levels. Although both graphs are necessary, comparison of only 3 groups as in the expression levels graph would increase its readability. Moreover, these graphs should be indicated with the letters (A and B) as mentioned in the text. Also, discussion regarding 9b graph is scarce (as well as the reason for choosing these two genes). If authors would like to keep all data in the graph representing the effects of melatonin supplementation, would be easier to read if presented in dots and trend lines format.

Despite these suggestions, authors should consider that showing the individual variations should be followed by a discussion on these differences, at least, mentioning it. I cannot find any information/discussion over individual variation visible in the 3 out of 9 graphs. Differences in the mean values, if not represents folds change, I would not consider as a significant individual variation.

Again, I would like to encourage the authors to simplify and generalize their finding or discuss individual variations showed in the graphs. In the present form, discussion refers to “groups”.

Author Response

Questionable is figure 9 where data of fertilisation capacity shows the same trend but individual variation is seen in the blastocyst BAX and BCL2L1 expression levels. Although both graphs are necessary, comparison of only 3 groups as in the expression levels graph would increase its readability. Moreover, these graphs should be indicated with the letters (A and B) as mentioned in the text. Also, discussion regarding 9b graph is scarce (as well as the reason for choosing these two genes). If authors would like to keep all data in the graph representing the effects of melatonin supplementation, would be easier to read if presented in dots and trend lines format.

Response: Thanks for the question. In order to make the manuscript more readable and clarity for readers, the BAX and BCL2L1 expression content had been deleted from the manuscript. This deletion would not affect the integrity and quality of the manuscript.

Despite these suggestions, authors should consider that showing the individual variations should be followed by a discussion on these differences, at least, mentioning it. I cannot find any information/discussion over individual variation visible in the 3 out of 9 graphs. Differences in the mean values, if not represents folds change, I would not consider as a significant individual variation.

Again, I would like to encourage the authors to simplify and generalize their finding or discuss individual variations showed in the graphs. In the present form, discussion refers to “groups”.

Response: Thanks for the question. The content has been added to display the individual variation, details in Line 349-352.

Reviewer 2 Report

The revised version of the manuscript drafted from Yang Li et al., provided a comprehensive analysis regarding the melatonin effectiveness on the fertilization capacity of sex sorted bull sperm. 

Authors ' conclude that melatonin treatment improves the fertilization capacity of sex sorted bull sperm mainly through apoptosis inhibition.

The introduction provides sufficient background and includes all relevant references. Research design is appropriate and methods employed are adequately described. Results are clearly presented and fully support the respective conclusions reported.

Authors revised the manuscipt appropriatelly according to Reviewer’s comments. 

Author Response

Thanks for the review. 

Round 3

Reviewer 1 Report

The authors response is not what I was asking for. Except providing in discussion information regarding individual variation in the blastocyst rate of melatonin-treated sperm (which refer to the IVF efficiency graph). There are 9 graphs where bulls are divided individually and I cannot find the reason behind this decision. The authors want to improve readability, but in my opinion, deleting the data difficult to discuss is not the proper way to do so. Again I ask the authors to consider:

1.     Merging the data from 3 bulls in graphs 1, 3, 4, 6, 7, 8 and 9 where individual differences did not influence the overall trend.

2.     Discuss the individual differences showed in graphs 2 and 5. To discuss these differences, statistical analysis between the corresponding results in these 3 males should be performed and added to the graph.

3.     When creating the graph, pay attention to provide statistical differences between the males if you want to show it and discuss it.

Author Response

The authors response is not what I was asking for. Except providing in discussion information regarding individual variation in the blastocyst rate of melatonin-treated sperm (which refer to the IVF efficiency graph). There are 9 graphs where bulls are divided individually and I cannot find the reason behind this decision. The authors want to improve readability, but in my opinion, deleting the data difficult to discuss is not the proper way to do so. Again I ask the authors to consider:

1.     Merging the data from 3 bulls in graphs 1, 3, 4, 6, 7, 8 and 9 where individual differences did not influence the overall trend.

Response: Thanks for the question. We have merged the data from 3 bulls according to the suggestion of reviewer.

2.     Discuss the individual differences showed in graphs 2 and 5. To discuss these differences, statistical analysis between the corresponding results in these 3 males should be performed and added to the graph.

Response: Thanks for the question. In order to make the results more clear and readable, we prefer to accept the comment that reviewer raised at the first time to merge the data from 3 bulls. Therefore, the content regarding the individual differences was deleted in the revised manuscript. We are sorry for this late correction, and we hope that the reviewer could forgive this for us.

3.     When creating the graph, pay attention to provide statistical differences between the males if you want to show it and discuss it.

Response: Thanks for the question. In order to make the results more clear and readable, we would like to accept the comment that reviewer raised at the first time to merge the data from 3 bulls. Therefore, the content regarding the individual differences was deleted in the revised manuscript. We are sorry for this late correction, and we hope that the reviewer could forgive this for us.